# Evaluation of the Binding Kinetics of RHEB with mTORC1 by In-Cell and In Vitro Assays

**DOI:** 10.3390/ijms22168766

**Published:** 2021-08-16

**Authors:** Raef Shams, Yoshihiro Ito, Hideyuki Miyatake

**Affiliations:** 1Emergent Bioengineering Materials Research Team, RIKEN Center for Emergent Matter Science, RIKEN, Wako 351-0198, Saitama, Japan; raef.shams@riken.jp (R.S.); y-ito@riken.jp (Y.I.); 2Department of Life Science, Graduate School of Science and Engineering, Saitama University, Saitama City 338-8570, Saitama, Japan; 3Nano Medical Engineering Laboratory, RIKEN Cluster for Pioneering Research, RIKEN, Wako 351-0198, Saitama, Japan

**Keywords:** mTORC1, RHEB, G-Protein, allosteric activation, kinase domain, binding kinetics

## Abstract

The mammalian/mechanistic target of rapamycin complex 1 (mTORC1) is activated by the small G-protein, Ras homolog enriched in brain (RHEB–GTPase). On lysosome, RHEB activates mTORC1 by binding the domains of N-heat, M-heat, and the focal adhesion targeting (FAT) domain, which allosterically regulates ATP binding in the active site for further phosphorylation. The crucial role of RHEB in regulating growth and survival through mTORC1 makes it a targetable site for anti-cancer therapeutics. However, the binding kinetics of RHEB to mTORC1 is still unknown at the molecular level. Therefore, we studied the kinetics by in vitro and in-cell protein–protein interaction (PPI) assays. To this end, we used the split-luciferase system (NanoBiT^®^) for in-cell studies and prepared proteins for the in vitro measurements. Consequently, we demonstrated that RHEB binds to the whole mTOR both in the presence or absence of GTPγS, with five-fold weaker affinity in the presence of GTPγS. In addition, RHEB bound to the truncated mTOR fragments of N-heat domain (∆N, aa 60–167) or M-heat domain (∆M, aa 967–1023) with the same affinity in the absence of GTP. The reconstructed binding site of RHEB, ∆N-FAT-M, however, bound to RHEB with the same affinity as ∆N-M, indicating that the FAT domain (∆FAT, aa 1240–1360) is dispensable for RHEB binding. Furthermore, RHEB bound to the truncated kinase domain (∆ATP, aa 2148–2300) with higher affinity than to ∆N-FAT-M. In conclusion, RHEB engages two different binding sites of mTOR, ∆N-FAT-M and ∆ATP, with higher affinity for ∆ATP, which likely regulates the kinase activity of mTOR through multiple different biding modes.

## 1. Introduction

The mammalian/mechanistic target of rapamycin (mTOR) regulates cell growth and survival through the modulation of the metabolic pathways [1,2]. mTOR assembles in two different complexes, mTOR complex 1 (mTORC1) and mTOR complex 2 (mTORC2), to regulate different processes [1,3]. In the mTORC1, mTOR is a kinase complexed with other proteins, regulatory associated protein of mTOR (Raptor), the mammalian lethal with SEC13 protein 8 (mLST8), DEP domain-containing mTOR-interacting protein (DEPTOR), and the 40-kDa proline-rich AKT substrate (PRAS40), which regulates the recruitment and phosphorylation of substrates (Figure 1a) [3,4]. In response to growth factors and nutrients, mTORC1 regulates a variety of life phenomena; synthesis of proteins, lipids and nucleotides, cell proliferation, and autophagy [1]. Recently, several studies have revealed the molecular mechanisms of mTORC1 kinase activation by amino acids and growth factors [5,6,7].

Early biochemical studies suggested that the small G-protein, Ras homolog enriched in brain (RHEB) was involved in the activation of mTORC1 [8,9,10]. In addition, it was found that the tuberous sclerosis complex 1/2 (TSC1/2), the upstream negative regulator of mTORC1, served as a GTPase-activating protein for RHEB [7,11,12]. Through this, the active RHEB–GTPase positively modulates the mTORC1 activity. Later, it was shown that RHEB binds to the ATP binding domain (aa 2148–2300) in a GTP-independent manner [10], which does not activate the kinase activity. Therefore, the functional aspect of the binding still has to be addressed [10].

It was reported that RHEB activates mTORC1 by antagonizing a negative regulator FKBP38, a member of FK506-binding protein family, in a GTP-dependent manner [8]. On the other hand, growth factors and nutrients promote the binding of the RHEB to mTOR, thereby promoting the kinase activity of mTORC1 [8]. Recently, the cryo-EM (cryogenic electron microscopy) structure of mTORC1/RHEB–GTPγS complex showed the mechanism of the mTORC1 activation by RHEB–GTP [4]. In the complex, RHEB–GTPγS bound to a binding site constituted by the N-heat, FAT, and M-heat domains far from the ATP binding site. It caused a large conformational change of mTOR, which in turn allosterically rearranged the ATP binding site to turn on the kinase activity [4]. Since RHEB is anchored to the lysosomal membrane mediated by farnesylation [12], the mTORC1 activation process occurs on the lysosome surface in response to growth factor and nutrient stimulation through two parallel and integrated pathways (Appendix A). At first, the RHEB–GTPase is being activated through the growth factor/TSC pathway, which enables RHEB to be charged by GTP [7,12]. Then, mTORC1 translocates onto the lysosome surface in response to the increasing concentration of amino acids [5,6]. The Raptor subunit of mTORC1 is anchored by the Rag GTPase-Ragulator complex onto the lysosome surface, offering the binding scaffold for RHEB–GTP [13,14]. On the lysosome surface, two RHEB–GTP complexes cooperatively activate mTORC1 to phosphorylate 4E-BP1 [4]. The stoichiometry obtained, however, did not involve the kinetics of RHEB binding to mTOR by means of protein–protein interaction (PPI) assays [15,16]. Therefore, in this study we aimed to reveal the kinetics of RHEB to mTOR, which can inform the development of new anti-cancer drugs.

## 2. Results and Discussion

Because mTORC1 and mTORC2 are often hyperactivated in cancer cells to sustain their rapid growth, its inhibition has been proposed for cancer therapy. Therefore, a variety of molecules have been developed to target the kinase activity of mTOR as anti-cancer agents [1,17,18,19]. However, specific inhibition of mTORC1 turned out to be more promising for cancer suppression than that of mTOR in both complexes. Thus, scientists have been seeking the ways to specifically target mTORC1 or, as a new strategy, to block signal transductions between mTORC1 and its regulatory proteins [11,20]. Accordingly, RHEB represents one of the potential targets for the specific inactivation of mTORC1 achieved by inhibiting the interaction between RHEB and mTORC1 [11]. Therefore, in this study, we aim to evaluate the binding kinetics of RHEB with mTOR to guide the development of new anti-cancer drugs.

At first, we studied the in vitro binding kinetics of RHEB to the whole mTOR. Briefly, the Halo-tagged full-length mTOR (aa 1–2,549; Kazusa-Promega, Appendix A [4]) was overexpressed by the pFN21A/HEK293 cell system, and purified by the HaloLink™ resin (Promega, Madison, WI, USA) (Figure 1b and Appendix A). The artificial gene of RHEB was synthesized (Eurofins Genomics, Tokyo, Japan) and subcloned into pET15b expression vector (Novagen, Merck Millipore, Darmstadt, Germany). 6xHis-tagged RHEB (aa 1–169, Appendix A [11]) was overexpressed in BL21(DE3) *E. coli* (Nippon gene, Toyama, Japan) and purified by Ni-NTA and Superdex-200 columns (GE Healthcare, Chicago, IL, USA) which was expressed as monomer/dimer mixture (Figure 1c and Appendix A). Next, we established the in vitro method for PPI determination by the AlphaLISA system including the anti-IgG donor beads and anti-6xHis acceptor beads (PerkinElmer, Waltham, MA, USA; Figure 1d and Appendix A). As a result, we observed that RHEB bound to mTOR in the presence or absence of GTPγS (Merck Millipore, Darmstadt, Germany), although only RHEB–GTP was shown to activate mTORC1 [4]. However, the binding affinity of RHEB–GTPγS to mTOR was five-fold weaker (K_D_ = 13.18 µM) than that of GTPγS-free RHEB (K_D_ = 2.44 µM; Figure 1e). This result suggests that a conformational change occurs upon GTP binding to RHEB, leading to the decreased binding affinity to mTOR. Because the GTP binding site is near the switch I of RHEB (aa 33–41), the binding of GTP probably interferes with the interaction between the switch I and mTOR domains involving M-heat and FAT [4].

The cryo-EM analysis revealed that RHEB interacted with three different mTOR fragments of aa 60–167 in N-heat domain (∆N), aa 967–1023 in M-heat domain (∆M), and aa 1240–1360 in FAT domain (∆FAT) (Figure 2a) [4]. On the other hand, RHEB is also reported to bind the fragment aa 2148–2300 of the ATP binding domain (∆ATP), which is far from the RHEB binding site involving N-heat, M-heat, and FAT domains (Figure 2a) [10]. To reveal the binding properties of the mTOR fragments, we assayed the binding of RHEB with ∆N, ∆M, ∆N-M (∆N + ∆M conjugates), ∆N-FAT-M (constructed to mimic the 3D arrangement of the fragments in the RHEB binding site where the FAT domain combines the N-heat and M-heat domains to organize the allosteric binding site [4]) or ∆ATP by the split-luciferase technology (NanoBiT^®^; Promega, Madison, WI, USA) (Appendix A) [21]. In the assay, the plasmids of RHEB–LgBiT and mTOR-SmBiT (representing different mTOR fragments) were co-transfected to HEK293 cells and incubated for 48 h. Then, the PPI determinants were assayed by measuring the luminescence intensity initiated by the addition of furimazine (Nano-Glo, Promega, Madison, WI, USA; Figure 2b). As a result, RHEB bound to all the mTOR fragments with different affinities irrespective to the endogenous GTP levels. The luminescence intensity of ∆N-M and ∆N-FAT-M fell into the same range, suggesting that ∆FAT was little involved in RHEB binding (Figure 2c). The single fragments of ∆N and ∆M showed a similar luminescence intensity, suggesting the same level of affinity for RHEB (Figure 2c). In the bindings, the mTOR fragment conjugates of ∆N-M or ∆N-FAT-M showed higher affinity than those of ∆N and ∆M, which suggested that the multiple fragments could increase the binding affinity in a cooperative manner (Figure 2c). In addition, ∆ATP showed the highest luminescence intensity, suggesting the strongest affinity for RHEB (Figure 2c). This result corresponds with the previous report that RHEB interacted with the ∆ATP domain [10]. Since the ∆ATP domain of mTOR (aa 2148–2300) is highly conserved in the PI3K family [10,22], it is possible that RHEB regulates the kinase activity of mTOR upon the binding to ∆ATP domain.

Based on the NanoBiT results, we further quantitatively measured the binding kinetics of RHEB with the mTOR fragments of ∆N-FAT-M, ∆N, and ∆ATP by the BLItz system (FortéBio, Fremont, CA, USA) [15]. For the measurements, we overexpressed the mTOR fragments by pET15b/BL21(DE3) *E. coli* system and purified them by Ni-NTA and Superdex-200 columns (GE Healthcare, USA) (Appendix A). The 6xHis-tag was then cleaved by thrombin from RHEB and further purified by His SpinTrap column (GE Healthcare, USA) to remove the cleaved 6xHis-taggs, and then by benzamidine column (GE Healthcare, USA) to remove thrombin. After that, 1 µM 6xHis-tagged mTOR fragments of ∆N-FAT-M, ∆N or ∆ATP were immobilized onto a Ni-NTA biosensor (FortéBio, Fremont, CA, USA) (Appendix A), and the 6xHis-tag cleaved-RHEB was used as analyte. As a result, RHEB interacted with the constructed allosteric binding site, ∆N-FAT-M, with K_D_ = 1.26 μM (Figure 3a–c, Table 1). On the other hand, ∆N showed a weaker affinity to RHEB with K_D_ = 6.47 μM (Figure 3d–f, Table 1), which corresponded to the in-cell results. Finally, RHEB bound to the ∆ATP with the highest affinity of K_D_ = 29 nM (Figure 3g–i, Table 1), as suggested by the in-cell assay (Figure 2c). These different binding affinities suggest the multiple functionalities of RHEB to regulate the kinase activity of mTORC1 [4]. We tried to measure the binding kinetics of the whole mTOR by the same method, through the immobilization of RHEB onto the Ni-NTA biosensor and using mTOR as analyte, but we could not figure it out due to the fast association/dissociation rates owing to the large molecular weight of mTOR complexes.

Overall, our study suggests that RHEB binds to mTORC1 in the presence or absence of GTP, suggesting multiple modes of RHEB to regulate mTORC1 activity. Previously, it has been reported that REHB activates mTORC1 in the presence of GTP [4,9,10,23]. In the activation, GTP changes the conformations of switch I and switch II regions of RHEB, respectively, to bind the RHEB binding site of mTORC1 [4]. In this study, however, we found that RHEB binds mTORC1 with higher affinity in the absence of GTP than that in the presence of GTP. We could resolve the inconsistency by assuming another binding site of mTORC1 for RHEB, even in the absence of GTP. The kinase domain of mTORC1 is reported to interact with RHEB [9,10], which suggests that the domain is involved in the RHEB binding in the absence of GTP. To confirm this, we measured the binding affinities of RHEB for the truncated kinase domain (∆ATP) and the fragmented allosteric binding site (∆N-FAT-M) in the absence of GTP, respectively. As a result, we found that RHEB binds to ∆ATP with much higher affinity than that of ∆N-FAT-M both in-cell and in vitro, which suggests that RHEB binds to the kinase domain of mTOR in the absence of GTP. Like the allosteric inhibition mode of rapamycin by binding to the kinase domain of mTORC1 with FKBP12 [24], RHEB may inhibit the kinase activity by posing a steric hindrance for the binding of ATP and/or the substrate proteins, providing a negative regulation mode for the kinase activity. Further studies should be conducted to confirm the scenario.

On the other hand, the FAT domain constructs the allosteric RHEB binding site, together with N-heat and M-heat domains [4]. However, our study showed that RHEB binds ∆N-FAT-M and ∆N-M fragments with the same affinity, suggesting little contribution of FAT domain for the RHEB binding. This information will contribute to designing a variety of peptide-based anti-cancer drugs to inhibit the kinase activity of mTORC1 by interfering with the mTORC1–RHEB interaction. The targeting of RHEB by compounds is emerging as a new modality for cancer therapy [11]. Therefore, this study will inform us to develop new inhibitors for mTORC1, based on the kinetics of RHEB obtained.

## 3. Materials and Methods

### 3.1. Materials

All the materials and softwares used in this study were listed in Table 2.

### 3.2. Methods

#### 3.2.1. mTOR Expression and Purification

Human mTOR ORF was supplied in pFN21A HaloTag^®^ CMV Flexi^®^ Vector (Kazusa Institute/Promega, Chiba, Japan), which was handled as previously described [25,26]. Briefly, HEK293 (RIKEN Cell Bank, RIKEN, Wako, Japan) cells were transfected by the Halo-tagged mTOR using FuGENE HD (Promega, Madison, WI, USA) transfection agent and grown as monolayer in 10 cm plates. Then, 10^8^ cells were collected and lysed on ice by 50 mM HEPES, pH 7.5, 150 mM NaCl, 1 mM EDTA, 1% (*v*/*v*) Triton X-100, 10% (*v*/*v*) glycerol, 0.1% (*w*/*v*) sodium deoxycholate, 0.005% (*v*/*v*) IGEPAL CA-630 buffer containing protease inhibitor cocktail tablets (cOmplete^TM^, Roche (Merck), Darmstadt, Germany), and 1.6 µg/mL DNase I. Cell lysate was then centrifuged at 13,200× *g* for 30 min at 4 °C, and the pellet was discarded. The supernatant was incubated with the pre-equilibrated HaloLink resin (Promega, Madison, WI, USA) for 1 h at RT on a rotator. Then, the resin was washed three times with the purification buffer. After that, mTOR was released from the resin by the TEV protease cleavage for further 1 h at RT on rotor. The released mTOR in the supernatant was carefully removed and incubated with Ni-NTA resin to remove the 6xHis-proteolytic TEV. Finally, the last supernatant containing mTOR was replaced by storage buffer (50 mM HEPES, pH 7.5, 10% (*v*/*v*) glycerol) and concentrated by the 30 kDa molecular weight cut-off (MW-CO) Amicon Ultra^®^ (Millipore (Merck), Darmstadt, Germany) to be stored at −80 °C. The yield was examined by western blotting using anti-mTOR antibody.

#### 3.2.2. Western Blotting

Western blotting was performed as previously described [27] to check mTOR expression and purification steps. After SDS–PAGE, the protein bands were transferred onto PVDF membrane (Millipore (Merck), Darmstadt, Germany) using Trans-Blot Transfer System (Bio-Rad, Hercules, CA, USA). Then, the membrane was blocked for 1 h at RT with 4% *w*/*v* skimmed milk 1xTBS-T, and after wash, it was incubated for 1 h at RT with primary antibody against mTOR, followed by the goat anti-rabbit secondary antibody incubation at RT or 2 h. Finally, the images of membranes were collected using the WSE-6100 LuminoGraph I (ATTO, Amherst, NY, USA). 

#### 3.2.3. Preparation of RHEB and Truncated mTOR Fragments

Plasmid construction

The proteins were prepared by the pET15b expression vector and BL21(DE3) *E. coli* (Nippon Gene, Toyama, Japan) [15,28]. Briefly, RHEB (507 bp), ΔN-FAT-M (858 bp), ΔN (294 bp), and ΔATP (459 bp) genes were cloned into pET15b expression vector using In-fusion cloning kit. The constructed plasmids were transformed into DH5α *E. coli* (Nippon Gene, Toyama, Japan) and spread over Luria–Bertani (LB) agar medium (Sigma-Aldrich, St. Louis, MO, USA) plates containing 0.1 mM ampicillin. Colony PCR and gene sequencing were performed to confirm the gene constructs and the plasmids were purified using NucleoSpin^®^ Plasmid EasyPure kit (Macherey-Nagel, Duren, Germany) according to the manufacturer’s protocol. Then, the purified plasmids were transformed into BL21(DE3) *E. coli* (Nippon Gene, Toyama, Japan) and incubated at 37 °C for 6 h in LB media containing 0.1 mM ampicillin. Protein expression was induced by 1 mM IPTG and further incubated overnight at 37 °C. SDS–PAGE showed that RHEB and ΔN were expressed as a soluble protein, while ΔN-FAT-M and ΔATP were expressed as inclusion bodies. 

b.Protein expression

At day 1, 5 liters terrific broth (TB) medium (Sigma-Aldrich, St. Louis, MO, USA) was prepared containing 0.8% (*v*/*v*) glycerol and autoclaved. Parallelly, 5 mL starter cultures were prepared as described above without IPTG and incubated overnight at 37 °C. At day 2, 0.1 mM ampicillin and 1% (*v*/*v*) antifoam silicon-type (SI) solution were added to the TB media and OD_600_ was checked as a reference. The starter cultures were added to the TB flasks and incubated at 37 °C with shaking (110 rpm). The OD_600_ values were measured hourly until they reached values ≥1.0, then 1 mM IPTG was added, and the culture was further incubated overnight. At day 3, cultures were centrifuged at 8000 rpm (15 min, 4 °C), and the cell pastes were stored at −80 °C.

c.Protein purification

Cell pastes (5 g for insoluble proteins or 10 g for soluble proteins) were resuspended in 100 mL lysis buffer (50 mM Tris-HCl, pH 8.0, 100 mM NaCl, 1 mM EDTA, 0.04 mg/mL lysozyme, 0.16 mg/mL DNase I) supplemented with protease inhibitor cocktail tablets (cOmplete^TM^, Roche (Merck), Darmstadt, Germany) and the suspensions were disrupted using a sonicator (Branson sonifier 250, Emerson Electric, St. Louis, MO, USA) on ice (5.0 W, 30–40% cycle/s, 5 min). For soluble proteins, cell lysate was ultra-centrifuged at 40,000 rpm for 1 h at 4 °C and the pellet was discarded. Then, the supernatant was applied to the HisTrapTM HP Ni-NTA column purified by ÄKTAprime plus fast protein liquid chromatography (FPLC) (GE Healthcare, Chicago, IL, USA) using binding buffer (50 mM Tris-HCl, pH 8.0, 10% (*v*/*v*) glycerol) and eluted by a gradient (0–100%) of elusion buffer (50 mM Tris-HCl, 10% (*v*/*v*) glycerol, 1 M imidazole, pH 8.0). The peak fractions were checked by SDS–PAGE, pooled, and loaded onto 3 kDa MW-CO Amicon Ultra^®^ (Millipore (Merck), Darmstadt, Germany) and washed with the binding buffer. Finally, proteins were concentrated and stored at −80 °C.

The recombinant ΔN-FAT-M and ΔATP have been expressed as inclusion bodies, so the purification process involved a refolding step. After cell disruption, the lysate was centrifuged at 8000 rpm for 5 min at 4 °C. Then, the pellet was washed 3 times by washing buffer (50 mM Tris-HCl, 100 mM NaCl, 1 mM EDTA, 4 M urea, pH 8.0) followed by 3 times washing by the same buffer without urea. The pellet was then resuspended in 20 mL solubilization buffer (50 mM Tris-HCl, 8 M urea, 100 mM NaH_2_PO_4_, 10 mM 2-mercaptoethanol, pH 8.0) on a rotator for 2 h at RT. After centrifugation (40,000 rpm, 1 h, 4 °C), the supernatant was diluted by the binding buffer (50 mM Tris-HCl, 8 M urea, 100 mM NaH_2_PO_4_, 5 mM 2-mercaptoethanol, pH 8.0) and loaded onto HisTrapTM FF crude Ni-NTA column (GE Healthcare, Chicago, IL, USA). After binding, the protein was eluted by a gradient (0–100%) of elusion buffer (50 mM Tris-HCl, 8 M urea, 100 mM NaH_2_PO_4_, 5 mM 2-mercaptoethanol, 1 M imidazole, pH 8.0). The peak fractions were checked by SDS–PAGE, pooled, and dropped into refolding buffer (50 mM Tris-HCl, 40 mM NaCl, 1 mM EDTA, 1 M L-arginine, and 10% (*v*/*v*) glycerol, pH 8.0). The refolded sample was then washed over a 3 kDa MW-CO filter (Millipore (Merck), Darmstadt, Germany) by the gel filtration buffer. After that, proteins were subjected to gel filtration using Superdex-200 column and running buffer of 50 mM Tris-HCl, pH 8.0, 150 mM NaCl, 10% (*v*/*v*) glycerol. After checking the purity on SDS–PAGE, the purified fractions were collected, concentrated by 3 kDa MW-CO Amicon Ultra^®^ (Millipore (Merck), Darmstadt, Germany), and the aliquots were stored at −80 °C.

#### 3.2.4. His-Tag Cleavage from RHEB

The RHEB aliquot containing 1 mg of RHEB was diluted 20 times in 1X PBS, and then mixed with 10 units of thrombin and incubated for 16 h at RT on a rotator to cleave the 6-His tag at the thrombin-cleavage site. Then, the solution was passed over the His SpinTrapTM column (GE Healthcare, Chicago, IL, USA) to remove 6-His tag which was eluted by 100, 200, and 500 mM imidazole in 1X PBS, respectively. To remove thrombin, the fractions were loaded onto HiTrapTM Benzamidine FF column (GE Healthcare, Chicago, IL, USA) equilibrated by binding buffer C (50 mM Tris-HCl, 100 mM NaCl, pH 8.0), and thrombin-free RHEB was gradually (0–100%) eluted by the elution buffer B (50 mM Tris-HCl, 500 mM NaCl, pH 8.0). 

#### 3.2.5. RHEB Charging with GTP*γ*S

We followed the RHEB charging protocol as previously described, with some modifications [11]. RHEB was incubated with 20-fold molar excess of GTP*γ*S (Millipore (Merck), Darmstadt, Germany) in the presence of 10 mM EDTA for 20 min at RT. Finally, the reaction was stopped by the addition of 20 mM of MgCl_2_. The yield was then passed over a PD SpinTrap G-25 column (GE Healthcare, Chicago, IL, USA) to remove excess GTP*γ*S.

#### 3.2.6. AlphaLISA Assay for RHEB–mTOR Protein–Protein Interaction (PPI)

A PPI assay based on energy transfer via donor/acceptor system using AlphaLISA^®^ assay (PerkinElmer, Waltham, MA, USA) was used [29]. Briefly, 100 µg/mL anti-IgG donor beads were incubated with excess anti-mTOR antibody for 1 h at RT followed by washing by 1× dilution buffer supplemented with the beads to remove unbound anti-mTOR antibodies. Then, a final concentration of 10 nM of purified mTOR was added to the donor beads solution and was further incubated for 1 h at RT. After the washes, the donor beads-mTOR complex was aliquoted (4 µL) in a 384-well OptiPlateTM (PerkinElmer, Waltham, MA, USA). Then, 6xHis–RHEB or 6xHis–RHEB–GTP*γ*S was titrated (3 µL of 0.1–100 µM) into the donor beads–mTOR mix and shaken gently. Finally, 3 µL of 200 µg/mL anti-6His acceptor beads were added to the mixture, the plate was top-sealed, covered, and incubated for more than 1 h at RT in dark. The same steps were performed to obtain baseline but without the addition of RHEB. The alpha signals were then measured by EnSpire^TM^ plate reader (PerkinElmer, Waltham, MA, USA). The data was normalized by subtracting the baseline values and the K_D_ values were calculated by the Prism software by using the specific binding model with the equation: Y = B_max_ (X/(Kd + X)), where Y, specific binding (measurement signal unit); B_max_, maximum binding (measurement signal unit); X, concentration of the analyte; Kd, the binding affinity.

#### 3.2.7. Preparation of Plasmids for In-Cell Protein–Protein Interaction (NanoBiT Assay)

RHEB (507 bp) gene was cloned into LgBiT vector, and ΔN-FAT-M (858 bp), ΔN-M (495 bp), ΔN (294 bp), ΔM (171 bp), and ΔATP (459 bp) genes were cloned into SmBiT vector using In-fusion cloning kit. The constructed plasmids were transformed into Dh5α *E. coli* and the cultures were spread over LB agar plates containing 0.1 mM ampicillin. Colony PCR and gene sequencing were performed to confirm the gene constructs, and the plasmids were purified using NucleoSpin^®^ Plasmid EasyPure kit (Macherey-Nagel, Duren, Germany) according to the manufacturer’s protocol. 

#### 3.2.8. In-Cell NanoBiT Assay

The NanoLuc Binary Technology (NanoBiT, Promega, Madison, WI, USA) based on split luciferase subunits can be used for the intracellular detection of PPI [21]. Briefly, 10^4^ HEK293 cells were seeded in DMEM (10% fetal bovine serum (FBS); 1% penicillin/streptomycin (P/S)) in B&W Isoplate-96 tissue-culture (TC) treated plates (PerkinElmer, Waltham, MA, USA) and incubated overnight (5% CO_2_; 37 °C). Then, the RHEB–LgBiT vector with the different SmBiT variants (50 ng/well each) were co-transfected into the cells using FuGENE HD at a ratio of 3:1 (*v*/*w*) and incubated for 48 h (5% CO_2_; 37 °C) for protein expression. Then, the culture medium was replaced by Opti-MEM reduced serum medium (Gibco, ThermoFisher, Waltham, MA, USA), and 20 µL of 20-fold diluted furimazine (Nano-Glo, Promega, Madison, WI, USA) was injected to initiate the luciferase reaction; the luminescence signal was measured using EnSpire plate reader (PerkinElmer, Waltham, MA, USA). 

#### 3.2.9. BLItz Measurements of RHEB Interactions with mTOR Truncates

We used the BLItz instrument (FortéBio, Fremont, CA, USA) to measure the binding kinetics of RHEB with the truncated mTOR fragment. At first, Ni-NTA biosensors (FortéBio, Fremont, CA, USA) were hydrated for 2 h in the kinetics buffer (10 mM HEPES, pH 7.4, 100 mM NaCl, 0.02% (*v*/*v*) Tween-20, and 5 mg/mL bovine serum albumin). In all measurements, the his-tagged truncated mTOR fragments (1 µM) were used as ligands to be immobilized onto the sensors, while tagless RHEB was used as analyte. In case of ∆N-FAT-M or ∆N, the measurement cycle composed of 30 s initial baseline (buffer), 120 s ligand loading, 120 s baseline (buffer), 180 s analyte association, and 300 s dissociation phases (buffer), while for ∆ATP, the cycle was shorter and divided into 30 s, 120 s, 60 s, 120 s, and 240 s, respectively. A reference cycle was applied for each sensor by introducing analyte only in the association phase to exclude nonspecific binding possibilities. RHEB concentrations were 0.5 and 1 µM ∆N-FAT-M or ∆N, and 0.05 and 0.1 µM for ∆ATP. All the experiments were performed at a shaking speed of 2000 rpm at 25 °C. Finally, the binding curves were fitted using 1:1 binding kinetics and analyzed by the BLItz Pro 1.2 software (FortéBio, Fremont, CA, USA) with the equations:

Association:Y = Y_0_ + R_eq_ (1−e^−Kobs × t^)

Dissociation:Y = Y_e_ + Y_∆_ e^−Kd × t^

Y, BLI signal in nm; Y_0_, the initial binding level; Y_e_, the fitted value of the exponential decay curve; Y_∆_, the nm shift difference between the first data point of the fitted dissociation curve and Y_e_; R_eq_, R equilibrium; K_obs_, the observed rate constant; K_d_, the dissociation rate constant; t, time in seconds.

#### 3.2.10. Data Analysis 

Statistical significance and number of samples are noted in the figure legends where appropriate. Data are expressed as mean ± SD. Ordinary one-way ANOVA was used as indicated; **** for *p* < 0.0001 and ns for *p* > 0.05. Statistical analyses were performed using GraphPad Prism software, v.8.4.3 (GraphPad, San Diego, CA, USA).

## 4. Conclusions

Although the previous studies showed that RHEB–GTP activated mTORC1 by several mechanisms, they did not reveal the binding kinetics [4,8,10]. Here, we studied the binding details of RHEB to whole mTOR and the truncated mTOR fragments [4,10]. In the assays, we used the in-cell and in vitro assays to facilitate the measurements [15,16]. RHEB bound to whole mTOR with 5 times weaker affinity in the presence of GTP than in its absence. On the other hand, the binding study of the truncated mTOR fragments involved in the reconstituted allosteric binding site suggested the cooperative binding mode of N-heat, M-heat, and FAT domains for RHEB. In addition, we observed that RHEB bound to the truncated ATP binding site in-cell and in vitro. The results show that the binding of RHEB to mTOR involves multiple binding sites with a variety of biding affinities, suggesting that RHEB regulates the kinase activity of mTOR through multiple mechanisms.

## Figures and Tables

**Figure 1 ijms-22-08766-f001:**
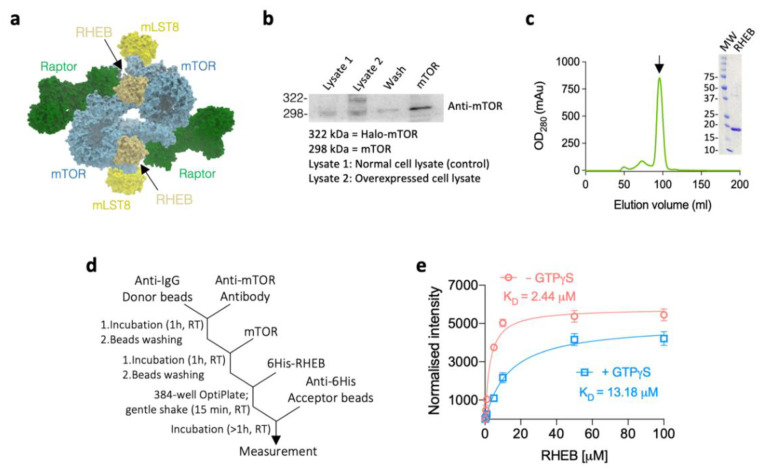
Binding kinetics of RHEB–mTOR. (**a**) Molecular structure of homodimeric mTORC1 showing the complex components except DEPTOR and PRAS40 (PDB ID: 6BCU) solved by the cryo-EM. (**b**) Western blot analysis of the different stages of mTOR purification showing the overexpression of the Halo-tagged mTOR. See the whole view of the blot in Appendix A. (**c**) Gel-filtration profile and the corresponding Coomassie brilliant blue R-250 stained SDS–PAGE (cropped from the full gel image in Appendix A) of the purified 6xHis–RHEB (*M*_W_: Molecular Weight; kDa). The arrow indicates the elution peak to be analyzed. The weak band above 37 kDa marker corresponds to the protein dimer. (**d**) Protocol of AlphaLISA^®^ assay to measure the binding affinity of RHEB for mTOR. Different concentrations of 6xHis–RHEB and excess amount of anti-6His acceptor beads were used (see Appendix A for details). RT, room temperature. (**e**) Binding of RHEB to mTOR in the presence (blue plot) or absence (red plot) of GTPγS. Data are shown as mean of two independent experiments (*n* = 3 replicates each) ± standard deviation (SD). The signal was normalized to the baseline. The equilibrium dissociation constants (K_D_) are shown.

**Figure 2 ijms-22-08766-f002:**
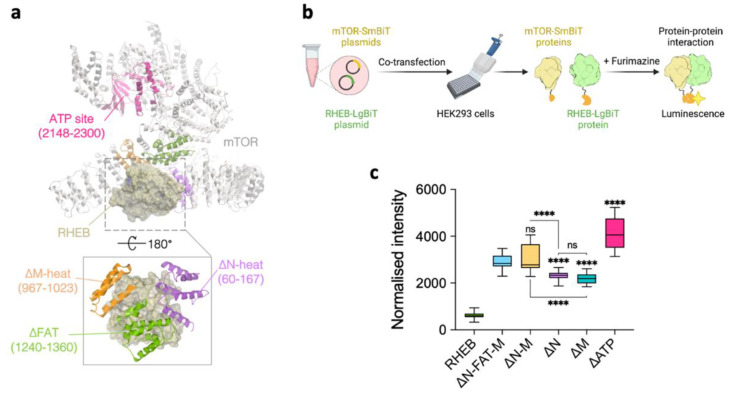
NanoBiT assay of RHEB binding to the different mTOR fragments. (**a**) The cryo-EM structure of RHEB complexed with mTOR (PDB ID: 6BCU) with the indicated mTOR fragments. (**b**,**c**) In-cell NanoBiT assay to evaluate the binding of the RHEB–LgBiT with the SmBiT-plasmids of the different mTOR fragments. (**b**) The plasmids were co-transfected to HEK293 cells, incubated for 48 h, and the luminescence reaction was initiated by furimazine addition (created by Biorender.com (accessed on 1 June 2021)). (**c**) Luminescence of the mTOR fragments with RHEB. Data are shown as mean of two independent experiments (*n* = 6 replicates each) ± SD. The signal was normalized to the background. Ordinary one-way ANOVA was used: **** *p* < 0.0001; ns, *p* > 0.05.

**Figure 3 ijms-22-08766-f003:**
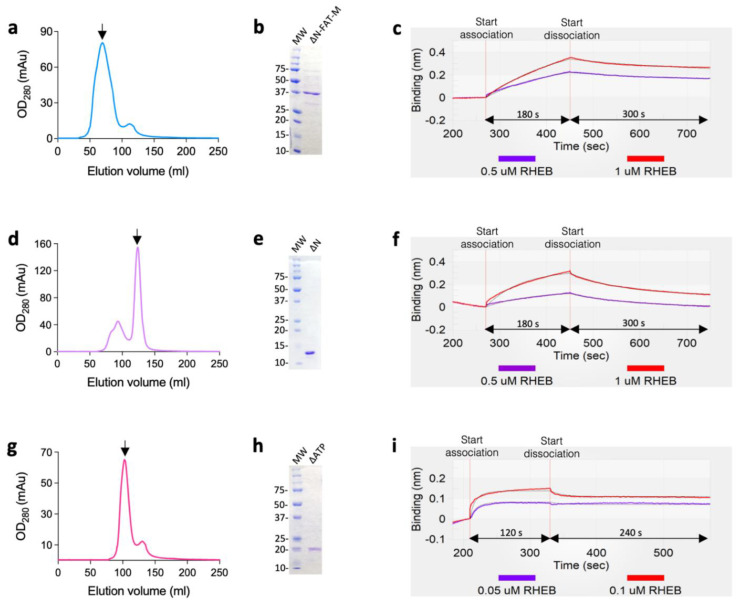
Binding kinetics of RHEB with mTOR fragments (1 µM) measured by BLItz. (**a**,**b**) Gel filtration profile (**a**) and the corresponding Coomassie brilliant blue R-250 stained SDS–PAGE (**b**) of ∆N-FAT-M. See Appendix A for full gel image. (**c**) Binding of RHEB to ΔN-FAT-M with association and dissociation phases of 180 s (270–450 s) and 300 s (450–750 s), respectively. (**d**,**e**) Gel filtration profile (**d**) and a representative Coomassie brilliant blue R-250 stained SDS–PAGE (**e**) of ∆N. See Appendix A for full gel image. (**f**) Binding of RHEB to ΔN with association and dissociation phases of (270–450 s) and 300 s (450–750 s), respectively. (**g**,**h**) Gel filtration profile (**g**) and a representative Coomassie brilliant blue R-250 stained SDS–PAGE (**h**) of ∆ATP. See Appendix A for full gel image. (**i**) Binding of RHEB to ΔATP with association and dissociation phases of 120 s (210–330 s) and 240 s (330–570 s), respectively. Global fitting was carried out for 1:1 binding kinetics. The calculated parameters are shown in Table 1. The arrows indicate the elution peaks used for the analysis.

**Table 1 ijms-22-08766-t001:** Kinetic parameters of 1:1 binding model of RHEB with the indicated proteins.

Protein	K_D_ (M) ^1,^*	k_a_ (M^−1^ s^−1^) ^2,^*	k_d_ (s^−1^) ^3,^*	χ^2 4^
∆N-FAT-M	1.26 ± 0.11 × 10^−6^	2.00 ± 0.12 × 10^3^	2.40 ± 0.11 × 10^−3^	0.04
∆N	6.47 ± 0.13 × 10^−6^	1.51 ± 0.30 × 10^3^	9.77 ± 0.07 × 10^−3^	0.04
∆ATP	2.91 ± 0.10 × 10^−8^	8.50 ± 0.13 × 10^5^	2.47 ± 0.03 × 10^−2^	0.02

^1^ K_D_, equilibrium dissociation constant. ^2^ k_a_, association rate constant. ^3^ k_d_, dissociation rate constant. ^4^ χ^2^, Chi-squared test of the fitted curve. * The values of K_D_, k_a_, and k_d_ are indicated ± the standard errors.

**Table 2 ijms-22-08766-t002:** Materials and softwares used in the study.

Reagent or Resource	Source	Identifier
**Antibodies**		
Rabbit mAb anti-mTOR	Cell Signaling Technology	2983; RRID: AB_2105622
Goat anti-rabbit-HRP secondary antibody	Invitrogen, Thermo Fisher	A16104; RRID: AB_2534776
**Bacterial Strains**		
ECOS^TM^ Competent *E. coli* DH5α	Nippon Gene	316-06233
ECOS^TM^ Competent *E. coli* BL21(DE3)	Nippon Gene	312-06534
**Chemicals and Recombinant Proteins**		
Dpn1 enzyme	Takara	1235A
ExoSAP-IT enzyme	Bioscience, Thermo Fisher	75001.1
KOD one PCR enzyme mix	Toyobo	KMM-201
DMEM (High-Glucose) media	FujiFilm Wako Pure Chemicals	044-29765
Opti-MEM media	Gibco, Thermo Fisher	31985-070
FuGENE HD	Promega	E2311
Ampicillin, sodium salt	Nacalai Tesque	02739-32
Kanamycin	FujiFilm Wako Pure Chemicals	113-00343
Isopropyl β-D-1-thiogalactopyranoside (IPTG)	Nacalai Tesque	19742-94
IGEPAL CA-630	MP Biomedicals	198596
Luria–Bertani agar media	Sigma-Aldrich	1002650948
Luria–Bertani Broth media	Nacalai Tesque	20068-75
Modified Terrific Broth media	Sigma-Aldrich	1002891164
Antifoam SI	FujiFilm Wako Pure Chemicals	018-17435
Protease inhibitors	Roche (Merck)	06538282001
Thrombin	FujiFilm Wako Pure Chemicals	206-18411
Coomassie brilliant blue R-250	FujiFilm Wako Pure Chemicals	6104-59-2
GTP*γ*S	Millipore	20-176
**Commercial kits**		
NucleoSpin^®^ EasyPure kit	Macherey-Nagel	740727.50
NanoBiT^®^ PPI Control Pair (FKBP/FRB)	Promega	N2016
Nano-Glo^®^ Live Cell Assay System	Promega	N2012
Anti-Rabbit IgG Alpha Donor beads	PerkinElmer	AS105M
Anti-6xHis AlphaLISA Acceptor beads	PerkinElmer	AL178M
HaloTag protein purification system	Promega	G6270
In-Fusion cloning kit	Takara	639650
HisTrap^TM^ HP Ni column	Cytiva	17524802
HisTrap^TM^ FF crude Ni column	Cytiva	17528601
Superdex-200 HiLoad 16/60 column	GE Healthcare	28-9893-35
His SpinTrap^TM^ column	GE Healthcare	28401353
HiTrap^TM^ Benzamidine FF column	GE Healthcare	17-5143-02
PD spintrap G-25	GE Healthcare	28918004
**Cell Lines**		
HEK293	RIKEN Cell Bank	N/A
**Oligonucleotides**		
Synthetic human RHEB gene (507 bp) UniportKB ID: Q15382	Eurofins Genomics	GSY1601-1
Human mTOR ORF/pFN21A UniportKB ID: P42345	Kazusa Institute / Promega	FHC01207
pET15b vector	Novagen	69661
**Primers for RHEB fragmentation for pET15b:** **FOR:** ′GTGCCGCGCGGGCAGCCAGTCCAAAAGCCGCAAAATC′ **REV:** ′ATCGATAAGCTTCTATTCCAACTTTTCCGCTTCCAG′	Eurofins Genomics	N/A
Primers for pET15b linearization: FOR: ′CATATGGCTGCCGCGCGGCACCAGGCCGCTGCTG′ REV: ′TAGAAGCTTATCGATGATAAGCTGTCAAACATGAG′	Eurofins Genomics	N/A
**Primers for RHEB fragmentation for LgBiT:** **FOR:** ′ATCGCCATGGTGGCCCAGTCCAAAAGCCGCAAAATC′ **REV:** ′ACTGCCTTGAGAAACTTCCAACTTTTCCGCTTCC′	Eurofins Genomics	N/A
**Primers for LgBiT vector linearization:** **FOR:** ′GTTTCTCAAGGCAGTTCAGGTGGTGGCGGGAGCGG′ **REV:** ′GGCCACCATGGCGATCGCTAGCGGTGGCTTTACC′	Eurofins Genomics	N/A
**Primers for SmBiT vector linearization:** **FOR:** ′TGGGCTAGCAGATCTTCTAGAGTCGGGGCGGCCGG′ **REV:** ′CATTCCACCGCTCGAGCCTCCACCTCCGCTCCCGC′	Eurofins Genomics	N/A
**Primers for mTOR^ΔN^ Fragment for SmBiT:** **FOR:** ′GGCTCGAGCGGTGGATCTACTCGCTTCTATGACC′ **REV:** ′AGAAGATCTGCTAGCACCCAGCCATTCCAGGGC′	Eurofins Genomics	N/A
**Primers for mTOR^ΔM^ Fragment for SmBiT:** **FOR:** ′GGCTCGAGCGGTGGACATCACACCATGGTTGTCC′ **REV:** ′AGAAGATCTGCTAGCCACAAAGGACACCAACATTC′	Eurofins Genomics	N/A
**Primers for mTOR^ΔN-M^ Fragment for SmBiT:** **FOR:** ′ACATGCACATCACACCATGGTTGTCCAGGCCATC′ **REV:** ′GTGTGATGTGCATGTCTCCGGCCCTCATTGCGG′	Eurofins Genomics	N/A
**Primers for mTOR^ΔN-F-M^ Fragment for SmBiT:** **FOR:** ′GGCCGGAGACATGCAGGCCAAGGGGATGCATTGG′ **REV:** ′AACCATGGTGTGATGCAAGTTTAAGAGGGTCTGTG′	Eurofins Genomics	N/A
**Primers for mTOR^ΔATP^ Fragment for SmBiT:** **FOR:** ′GCTCGAGCGGTGGACAGCCAATCATTCGCATTCAG′ **REV:** ′AGAAGATCTGCTAGCGGCCAGGTCGTCCCCAGCTG′	Eurofins Genomics	N/A
**Primers for mTOR^ΔN^ Fragment for pET15b:** **FOR:** ′GTGCCGCGCGGCAGCTCTACTCGCTTCTATGACC′ **REV:** ′ATCGATAAGCTTCTAACCCAGCCATTCCAGGGCTC′	Eurofins Genomics	N/A
**Primers for mTOR^ΔN-F-M^ Fragment for pET15b:** **FOR:** ′GATAACGCGATCGCCTCTACTCGCTTCTATGACC′ **REV:** ′CGAATTCGTTTAAACCACAAAGGACACCAACATTC′	Eurofins Genomics	N/A
**Primers for mTOR^ΔATP^ Fragment for pET15b:** **FOR:** ′GATAACGCGATCGCCCTGCCTCAGCTCACATCC′ **REV:** ′CGAATTCGTTTAAACGCATGTGATTCTGTAGTTGC′	Eurofins Genomics	N/A
**Primers for colony PCR of SmBiT/LgBiT:** **FOR:** ′GAAGTCGAACACGCAGATGCAGTCG′ **REV:** ′CACTGCATTCTAGTTGTGGTTTGTCCAAACTC′	Eurofins Genomics	N/A
**Primers for colony PCR of pET15b:** **FOR:** ′CGATCCCGCGAAATTAATACGACTCACTATAG′ **REV:** ′GACATTAACCTATAAAAATAGGCGTATCACGAGG′	Eurofins Genomics	N/A
**Software**		
ICM-Pro 3.9 software	Molsoft L.L.C.	https://www.molsoft.com/products.html
SnapGene 5.1.7 software	GSL Biotech L.L.C.	https://www.snapgene.com/
BLItzPro 1.2 software	FortéBio (Sartorius)	https://www.sartorius.com/
Prism 8.4.3 software	GraphPad	https://www.graphpad.com/

## Data Availability

Not applicable.

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
