# Peer review of "Evaluation of the Binding Kinetics of RHEB with mTORC1 by In-Cell and In Vitro Assays"

_ijms, 2021, doi:10.3390/ijms22168766_

Round 1

Reviewer 1 Report

Esteemed Authors,

It has been a great honor, as well as a pleasantly challenging activity, to review the article entitled Evaluation of the Binding Kinetics of RHEB with mTORC1 by In-cell and In vitro Assays.

The topic addressed in the article is of particular importance for medicine and mainly for cell growth and proliferation, immunomodulation, synthesis of proteins, lipids, nucleotides, and inhibition processes. Sirolimus, also known as rapamycin, is a macrolide compound used to coat coronary stents, prevent organ transplant rejection and treat a rare lung disease called lymphangioleiomyomatosis. It has immunosuppressant functions in humans and is especially useful in preventing the rejection of kidney transplants.

The article is structured following the classic model for this type of material (Communication) and comprises four parts: Introduction, Results and Discussion, Materials and Methods, and Conclusion. All the four major components of the article are balanced dimension-wise and presented coherently and logically, tightly linked.

However, some issues need to be corrected. The first refers to the order of the chapters: as a rule, the chapter reserved for materials and methods precedes the chapter dedicated for results. A second problem concerns the numbering of chapters, from 1 to 4 and not from 1 to 5, as presented in the original version of the article.

With some minor exceptions, the materials and methods are specified and described adequately. All iconographic documents – two tables and three figures - were given accurate descriptions, the results were described in great detail, and the conclusions are adequate.

The documentation is adequate, and all the authors are cited in the text of the paper, without exception.

The provided scientific results are exact and precise. The goal of the conducted research is well specified and delineated. The working protocol is appropriate, and the used analysis methods are correlated with the proposed objectives.

As for the grammar of the paper, the article is very well written: only a few suggestions in the grammar of the text can be mentioned, as follows:

Page 1, line 25 – replace “regulates the” with “regulate the”;

Page 3, line 82 – replace “decrease of” with “decrease in”;

Page 4, line 131 – replace “different bindings obtained suggests” with “different obtained bindings suggest”;

Page 7, line 150 – replace “described1, 2.” with “described.”;

Page 7, line 162 – replace “described 3 to” with “described to”;

Page 7, line 170 – replace “4, 5 Briefly, RHEB” with “Briefly, RHEB”;

Page 8, line 211 – replace “was prepared” with “were prepared”;

Page 8, line 218 – replace “modifications6” with “modifications”;

Page 9, line 244 – replace “used7” with “used”;

Page 9, line 248 – replace “was hydrated” with “were hydrated”.

As a general conclusion regarding the grammar, the text does not contains other mistakes that need to be corrected. As for the editing (writing), the article should be checked carefully as there are some minor typos, especially in the text of the paper and less in the list of bibliographic references.

The presence at the end of the article of a list of abbreviations is another positive element, which simplifies the text, makes it shorter and easier to navigate.

Together with other positive elements, the presentation's scientific relevance and quality will surely make the article attractive to a broad audience, especially to the authors interested in medicine, biochemistry, molecular biology, metabolism, and public health.

Provided that the authors revise the material and improve on the elements mentioned above, the paper may be accepted and published in the International Journal of Molecular Sciences.

            Best Regards,

            Reviewer

Reviewer 2 Report

Major point:

The study lacks discussion. Please highlight the significance of the newly acquired findings in the light of previous research as part of a Discussion section. It is for example not clear how the revealed kinetic data of the interaction between RHEB and mTORC1 may contribute to informing cancer therapy?

Minor points:

1) The name of the supplementary file "ijms-1274894-non-published" is confusing as it suggests that the supplementary data will not be published. Please rename this file to better indicate a supplementary material.

2) It may be useful to include a schematic of primary structures of the various mTOR fragment domains used aligned below the full-length sequence of mTOR as part of a new figure panel or new figure.

3) Please replace "(R. S.)" with "(R.S.)" (line 6 and supplementary file affiliations).

4) Please change "(Y. I)" to "(Y.I.)" (line 6 and supplementary file affiliations).

5) Please replace "(H. M.)" with "(H.M.)" (line 8 and supplementary file affiliations).

6) Although FAT domain is mentioned in the beginning of the Abstract, it is not mentioned later in the Abstract in terms of the experimental outcome. Please comment on the ability of FAT domain to interact with RHEB in the Abstract.

7) Similarly, the ATP binding site is mentioned only in the Abstract beginning, but the significance of RHEB binding to the truncated ATP binding site domain is missing from the conclusion part of the Abstract.

8) Please define abbreviation for "FAT" (line 14), "cryo-EM" (line 49), "RT" (Figure 1d), "LB" (line 172), "SI" (line 182), "MW-CO" (line 206), "FBS" (line 240), "P/S" (line 240), "TC" (line 240), "SD" (line 259) and "FPLC" (legend to Figure S3b) at the site of first occurrence as well as in the Abbreviations section.

9) "For this" could be changed to "To this end" (line 18).

10) Please replace "studies, and" to "studies and" (line 19).

11) Please change "it was shown" to "we demonstrate" (line 19).

12) Although the authors claim that RHEB binds to the N-heat and M-heat domain in a GTP-independent manner in "Also, RHEB bound to the truncated mTOR fragments of N-heat domain (60-167) and M-heat domain (967-1023) in a GTP independent manner" (line 21), the effect of GTP on RHEB and N-heat or M-heat domain association has not been tested. Please correct.

13) Please replace "60-167" with "aa 60-167" (lines 22, 85, 101)?

14) Please change "967-1023" to "aa 967-1023" (lines 22, 86)?

15) Please replace "2148-2300" to "aa 2148-2300" (lines 23, 44, 87, 101)?

16) Please replace "GTP independent" to "GTP-independent" (line 22).

17) It is not clear what the authors mean by "higher affinity also in GTP independently" in "Furthermore, RHEB bound to the truncated kinase domain (2148-2300) with higher affinity also in GTP independently" (line 22)?

18) The statement "are involved in different upstream or downstream signals" does not seem to make sense (line 32).

19) Please change "phenomenon" to "phenomena" (line 35).

20) Please replace "regulators" with "regulator" (line 41).

21) Please change "bound" to "binds" (lines 44).

22) Please replace "in GTP-independent" with "in a GTP-independent" (line 44).

23) Please change "did" to "does" (line 45).

24) Please replace "activated" with "activates" (line 47).

25) Please change "whereby" to "thereby" (line 49).

26) Please replace "a scenario how mTORC1 was activated" with "the mechanism of mTORC1 activation" (line 50).

27) "farnesylated into the lysosome membrane" (line 53) might not be correct. In strict terms, protein can be farnesylated or anchored to a membrane but not "farnesylated into a membrane". Please rephrase.

28) Please replace "onto" to "on" (line 54).

29) Please change "growth factors / TSC" to "growth factor/TSC" (line 55).

30) Please replace "was translocated" to "translocates" (line 56).

31) Please change "was" to "is" (line 57).

32) Please replace "activated" with "activate" (line 59).

33) Please change "the RHEB" to "RHEB binding" (line 61).

34) Please replace "by pFN21A/ HEK239" with "in pFN21A/HEK239" (line 64).

35) Please change "(Figure S2 and Figure 1b)" to "(Figure 1b and Figure S2)" (line 65).

36) Please replace "by" with "in" (line 66).

37) Please change "(Figure S3 and Figure 1c)" to "(Figure 1c and Figure S3)" (line 68).

38) Please replace "PPI" with something like "PPI determination" or ""PPI affinity determination" (line 68).

39) Please indicate whether bands from Figure 1b are identical to those shown in Figure S2b in the respective figure legend.

40) From Figure 1c is not clear which peak corresponds to the fractions collected for Western blot quantification depicted in Figure S3d?

41) Despite there is a faint band below the 37 kDa molecular weight marker in Figure 1c this band appears above the same marker in Figure S3d. Please explain this discrepancy in the text.

42) It is not clear for how long and at what temperature was OptiPlate shaken in Figure 1d?

43) Please change "Deptor or" to "DEPTOR and" (line 72).

44) Please replace "Coomassie blue-stained" with "Coomassie brilliant blue-stained" (line 73, legend to Figures S3c, S3d, S6c, S6d, S7c, S7d, S8c, S8d).

45) Please change "KDa" to "kDa" (lines 74, 194, abbreviation list - FKBP12 and PRAS40, legend to Figure S3c, S6c, S7c, S8c).

46) Please replace "A variety concentration" with "Different concentrations" (line 75).

47) Please change "See Figure S2" to "see Figure S4" (line 75).

48) Please replace "Data shown" with "Data are shown" (line 76).

49) The authors claim that "The signal was normalized to the background" (line 77) in Figure 1e, however normalizing data to a background noise does not make sense in science. Do the authors actually mean that the signal was normalized to the baseline? If yes, specify exactly this baseline in the text. If not, please replot all data points without normalizing to the background noise. Also, please specify equation used to fit these data sets to derive the dissociation constant. This can be done in the Materials and Methods section.

50) Please replace "mTORC1[4]" with "mTORC1 [4]" (line 80).

51) "happens" could better read as "occurs" (line 82).

52) Please change "decrease of the" to "decreased" (line 82).

53) Please replace "33-41" with "aa 33-41" (line 83)?

54) Please change "interferes the" to "interferes with the" (line 83).

55) Whereas the authors claim that ΔFAT is composed of 1240-1360 amino acid sequence in "The cryo-EM analysis revealed that RHEB interacted with three different mTOR fragments of 60-167 in N-heat domain (hereafter termed as ΔN), 967-1023 in M-heat domain (ΔM) and 1240-1360 in FAT domain (ΔFAT)" (line 85) and in "Construction and purification of overexpressed ∆N-FAT-M (60-167, 1240-1360, 967-1023)" (Figure S6 legend), Figure 2 actually shows 1277-1307 for ΔFAT. Please correct this dichotomy.

56) Please replace "1240-1360" with "aa 1240-1360" (line 86).

57) "the plasmids of RHEB-LgBiT and each mTOR fragment-SmBiT" (line 91) seems not to be grammatically correct. Please revise.

58) Please change "was" to "were" (line 92).

59) Please replace "luciferin (furimazine)" with "furimazine" (line 92) as these are different substrates.

60) The authors claim that furimazine was used as a luminiscent substrate in "Then, the PPI was assayed by measuring the luminescence intensity initiated by the addition of luciferin (furimazine)" (line 92), this is indicated as "luciferin" in Figure 2b and its legend. Please provide unambiguous description of the substrate used.

61) It is not clear what the authors mean by "For the analysis, we normalized the GTP concentration considering of the inherent GTP in cells" (line 93) with respect to "we normalized the GTP concentration" and "inherent GTP in cells"? How does GTP concentration relate to the NanoBiT assay and why there is a need to use its normalized value? Do the authors actually mean endogenous GTP levels instead of "inherent GTP in cells"?

62) It is not clear why GTP concentration normalization should affect ("As a result") RHEB binding to mTOR fragments in "As a result, RHEB bound to all the mTOR fragments but with different affinities" (line 94)?

63) Please change "considering of the" to "considering the" (line 93).

64) Please replace "a little involved in the" with "little involved in" (line 95).

65) "On the other hand, the single fragments of ΔN and ΔM showed a similar luminescence intensity, suggesting a same level of affinity" (line 96) does not semantically well follow "The luminescence intensity of ΔN-M and ΔN-FAT-M fell into the same range, suggesting that ΔFAT was a little involved in the RHEB binding" (line 94) with respect to "On the other hand" since the observed luminiscence was actually similar in both instances ("luminescence intensity of ΔN-M and ΔN-FAT-M fell into the same range" and "ΔN and ΔM showed a similar luminescence intensity") thereby precluding the use of "On the other hand" to connect these sentences.

66) Please replace "cooperation" with "cooperative" (line 99).

67) Please change "indication of mTOR fragments" to "indicated mTOR fragments" (line 116).

68) Please replace "Data shown as mean ± standard deviation from a representative result of two independent experiments (n=6 replicates each)" with "Data are shown as mean of two independent experiments (n=6 replicates each) ± standard deviation" (line 119).

69) Please replace "As results" with "In result" (line 128).

70) "bindings" could be changed to "binding affinities" (line 131).

71) "but we could not be due to the fast association/dissociation rates because of the large molecular weight of mTOR complexes" (line 134) is not grammatically correct with respect to "we could not be due to the fast association/dissociation rates because". Please rephrase.

72) From Figure 3a is not clear which peak corresponds to the fractions collected for Western blot quantification depicted in Figure S6d?

73) Please indicate whether bands from Figure 3b are identical to those shown in Figure S6d in the respective figure legend.

74) The authors claim that "Binding of RHEB to ΔN-FAT-M with association and dissociation phases at 180 s and 300 s, respectively" (line 138), however the association and dissociation times differ from those shown in Figure 3c. Please revise.

75) It is not clear what do the red vertical lines indicate in Figures 3c, 3f, and 3i? Please comment on in the respective figure legends and explain exactly how were the association and dissociation events triggered? Also, please specify equation used to fit these data sets to derive the kinetic parameters shown in Table 1. This can be done in the Materials and Methods section.

76) From Figure 3d is not clear which peak corresponds to the fractions collected for Western blot quantification depicted in Figure S7d?

77) The authors claim that "Binding of RHEB to ΔN with association and dissociation phases at 180 s and 300 s, respectively" (line 140), however the association and dissociation times differ from those shown in Figure 3f. Please fix.

78) From Figure 3g is not clear which peak corresponds to the fractions collected for Western blot quantification depicted in Figure S8d?

79) The authors claim that "Binding of RHEB to ΔATP with association and dissociation phases at 120 s and 240 s, respectively" (line 141), however the association and dissociation times differ from those shown in Figure 3i. Please correct.

80) Please format the "Source" column as part of the Materials and Methods section so that all entries have a uniform font size.

81) Please format "https://www.sartorius.com/" in the "Identifier" column as part of the Materials and Methods section using a uniform font size.

82) Please convert "1, 2" to a proper citation (line 150).

83) Please replace "10-cm" with "10 cm" (line 151).

84) It is not clear what the authors mean by "(108)" in "Then, the cells (108) were collected and lysed on ice by 50 mM HEPES, pH 7.5, 150 mM NaCl, 1 mM EDTA, 1%(v/v) Triton X-100, 10 %(v/v) glycerol, 0.1%(w/v) sodium deoxycholate, 0.005%(v/v) IGEPAL buffer containing protease inhibitors and 1.6 μg/ml DNase I" (line 151)?

85) It is not clear what protease inhibitors were used as part of the IGEPAL buffer used to lyse the cells in "Then, the cells (108) were collected and lysed on ice by 50 mM HEPES, pH 7.5, 150 mM NaCl, 1 mM EDTA, 1%(v/v) Triton X-100, 10 %(v/v) glycerol, 0.1%(w/v) sodium deoxycholate, 0.005%(v/v) IGEPAL buffer containing protease inhibitors and 1.6 μg/ml DNase I" (line 151)?

86) Please change "1%(v/v) Triton X-100, 10 %(v/v) glycerol, 0.1%(w/v) sodium deoxycholate, 0.005%(v/v)" to "1% (v/v) Triton X-100, 10% (v/v) glycerol, 0.1% (w/v) sodium deoxycholate, 0.005% (v/v)" (line 152).

87) Please specify the type of IGEPAL used (line 153). Was this IGEPAL CA-630?

88) Please replace "4°C" with "4 °C" (lines 154, 201).

89) It is not clear how exactly was the final supernatant concentrated in "Finally, the last supernatant containing mTOR was replaced by storage buffer (50 mM HEPES, pH 7.5, 10 %(v/v) glycerol) and concentrated to be stored at -80 °C" (line 158)?

90) Please convert "3" to a proper citation (line 162).

91) Please change "1× TBS-T" to "1×TBS-T" (line 164).

92) Please replace "RHEB and mTOR truncates" with "truncated RHEB and mTOR" (line 168).

93) Please convert "4, 5" to a proper citation (line 170).

94) Please replace "genes constructions, and" with "gene constructs and" (lines 173, 236).

95) Please change "manufacturer" to "manufacturer's" (lines 174, 237).

96) Please change "37°C" to "37 °C" (lines 175, 176, 181, 241).

97) Please replace "The proteins expressions were" with "Protein expression was" (line 176).

98) Please change "body" to "bodies" (line 178).

99) Please replace "5 L Terrific broth media, TB" with "5 l of Terrific broth (TB) medium" (line 180).

100) Please replace "till reached" with "till they reached values" (line 184).

101) Please change "and further incubated for overnight was followed" to "and the culture was further incubated overnight" (line 184).

102) Please replace "cultures solutions" with "cultures" (line 185).

103) Please change "cell" to "the cell" (line 185).

104) Please replace "-80°C" with "-80 °C" (line 185).

105) It is not clear what protease inhibitor was used in "Cell pasts (5 g for insoluble proteins or 10 g for soluble proteins) were resuspended in 100 ml lysis buffer (50 mM Tris-HCl, pH 8.0, 100 mM NaCl, 1mM EDTA, 0.04 mg/ml lysozyme, 0.16 mg/ml DNase I) supplemented with 1x protease inhibitor and the suspension was disrupted using cell disruptor (5.0 W, 30-40 % cycle / sec, 5 min) on ice" (line 187)?

106) Please change "pasts" with "pastes" (line 187).

107) Please replace "1mM" with "1 mM" (line 188).

108) Please change "1h" with "1 h" (lines 190, 230).

109) Please change "10 % v/v" to "10% (v/v)" (lines 192, 193, 205, 208).

110) Please replace "elusion" with "of elusion" (lines 192, 203).

111) Please change "for washing by" with "and washed with" (line 194).

112) Please replace "had" with "have" (line 196).

113) Please change "other 3 times" to "3 times" (line 199).

114) Please format "2" and "4" in "NaH2PO4" using lowercase capitalization (lines 200, 202, 204).

115) Please replace "2-mercatoethanol" with "2-mercaptoethanol" (lines 200, 202, 204).

116) Please change "3KDa" to "3 kDa" (line 206).

117) Please replace "for" with "to" (line 207).

118) Please change "using running" with "and running" (line 207).

119) It is not clear exactly how were the fractions concentrated in "After SDS-PAGE purity confirmation, the purified fractions were collected, concentrated, and stored at -80 °C" (line 208)?

120) Please replace "SDS-PAGE purity confirmation" with "checking the purity on SDS-PAGE" (line 208).

121) It is not clear what the authors mean by "A ratio of 1 mg protein in 20 times 1X PBS" in "A ratio of 1 mg protein in 20 times 1X PBS: 10 units of thrombin was prepared and incubated for 16 h at room temperature on rotor to cleave the 6-His tag at the thrombin site" (line 211)?

122) Please change "which eluted" to "which was eluted" (line 213).

123) Please replace "100-, 200- and 500-mM" with "100, 200, and 500 mM" (line 213).

124) Please change "Following the fractionation and to remove thrombin, the solution was loaded" to "To remove thrombin, the solution was loaded, following the fractionation," (line 213).

125) Please replace "to be" with "and" (line 215).

126) Please convert "6" to a proper citation (line 218).

127) Please format "2" in "MgCl2" using lowercase capitalization (line 220).

128) Please change "protein: protein" to "protein:protein" (lines 222, 223, 232 and the Abbreviations section (PPI)).

129) Please replace "protein: protein interaction (PPI)" with "PPI" (line 223).

130) Please convert "7" to a proper citation (line 224).

131) Please change "was" to "were" (lines 224, 248).

132) Please replace "beads" with "the" (line 227).

133) Please change "e." to "E." (line 235).

134) Please replace "and spread" with "and the cultures were spread" (line 235).

135) Please change "Luciferase" to "luciferase" (line 239).

136) Please convert "8" to a proper citation (line 240).

137) Please change "D-MEM" to "DMEM" (line 240 and Materials and Methods section (Chemicals and Recombinant Proteins)).

138) Please format "4" in "104" using uppercase capitalization (line 240).

139) Please format "2" in "CO2" using lowercase capitalization (lines 241, 243).

140) Please replace ", and" with "and" (line 243).

141) Please change "20×" to "20-fold" (line 244).

142) Please replace "luciferase interaction" with "luciferase reaction interaction" (line 244).

143) Please change "v/v Tween-20" to "(v/v) Tween 20" (line 249).

144) Please replace "30s" with "30 s" (lines 251, 253).

145) Please change "120s" to "120 s" (lines 251, 252, 253).

146) Please replace "180s" with "180 s" (line 252).

147) Please change "300s" to "300 s" (line 252).

148) Please replace "60s, 120s and 240s" with "60 s, 120 s, and 240 s" (line 253).

149) Please change "indicated; **** for P < 0.0001, *** for P < 0.001, ** for P < 259 0.01, * for P < 0.05, and ns for P > 0.05" to "indicated; **** for P < 0.0001 and ns for P > 0.05" (line 253).

150) Please replace "that of GTP" with "in its" (line 265).

151) Please change "mechanism" to "mechanisms" (line 269).

152) It is not clear what does "(C)" mean in "R.S. was supported by the Junior Research Associate (JRA) Program in RIKEN. H.M. was partly supported by the Incentive Research Program in RIKEN (FY2018-2019, FY2019-2020) and JSPS Grant-in-Aid for Scientific Research (C) (JP20K06516)" (line 279).

153) It is not clear what does "Bio-material Analysis" refer to (line 281)?

154) Please format "(https://www.editage.jp/)" using a standard font size (line 282).

155) Please replace "12-KDa" with "12 kDa" in the Abbreviations section (FKBP12).

156) Please change "Binary Technology" to "binary technology" in the Abbreviations section (LgBiT and SmBiT).

157) Please replace "Mammalian/Mechanistic" to "Mammalian/mechanistic" in the Abbreviations section (mTOR).

158) Please change "Phosphoinositide 3-kinases" to "Phosphoinositide 3-kinase" in the Abbreviations section (PI3K).

159) Please replace "Protein: protein interaction" to "Protein:protein interaction" in the Abbreviations section (PPI).

160) Please change "KDa" to "kDa" in the Abbreviations section (PRAS40).

161) It is not clear what the authors mean by "TEV site" in Figure S2a?

162) It is not clear at which step was TEV cleavage performed in Figure S2a?

163) It is not clear how was mTOR eluted from the HaloLink resin in Figure S2a?

164) It is not clear at which step was thrombin cleavage performed in Figure S3a?

165) It is not clear how was RHEB eluted from the Ni-NTA and Superdex-200 columns in Figure S3a?

166) The label "Manual Run 0:13_UV" is very difficult to read in Figure S3b. Please increase font size or delete this label.

167) Both x and y axis descriptions are very difficult to read in Figure S3b. Please increase font size.

168) It is not clear what the authors mean by "Fractions peak" in Figure S3b. Are these the fractions containing the eluted RHEB?

169) It is not clear what are the other two dominant peaks appearing between around 100 and 225 ml in the chromatogram shown in Figure S3b?

170) In addition to 10, 15, 20, and 25 sizes, please annotate all other molecular weight markers shown in the Coomassie brilliant blue-stained gels present in Figures S3c and S3d.

171) It is not clear what fractions depicted in Figure S3c were used for downstream purification on Superdex-200?

172) It is not clear what are the contaminating bands appearing in Figure S3c? Please comment on in the figure legend.

173) It is not clear what fractions depicted in Figure S3d were used as the final purification product of RHEB?

174) It is not clear what sample was loaded in the rightmost lane in Figure S3d as it contains an extra higher molecular weight band? This band seems to be present also in the leftmost fraction but with lower intensity. Please comment on in the figure legend.

175) It is not clear which lanes are denoted as fractions in Figures S3a, S3d, S6a, S6d, S7a, S7d, S8a, S8d as the horizontal line indicator does not exactly coincide with lanes underneath.

176) Please change "superdex-200" to "Superdex-200" in the legend to Figures S3a, S3d, S6a, S6d, S7a, S7d, S8a, S8d.

177) It is not clear what type of Coomassie blue was used in Figures S3c, S3d, S6c, S6d, S7c, S7d, S8c, and S8d? Was this G-250 or R-250? Please specify this parameter in the respective figure legends.

178) The role of "1/2O2" in the schematics illustrated in Figure S4 (bottom picture) is not clear. Please explain in the figure legend.

179) Please replace "method to determine" with "method used to determine" in the legend to Figure S4.

180) It is not clear what does (GGS)5 mean in Figure S5? Please explain in the figure legend.

181) Please change "N terminal" to "N-terminus" in the legend to Figure S5.

182) Please change "C terminal" to "C-terminus" in the legend to Figure S5.

183) It is not clear at which step was thrombin cleavage performed in Figure S6a?

184) It is not clear how was ∆N-FAT-M eluted from the Ni-NTA and Superdex-200 columns in Figure S6a?

185) The label "Manual Run 0:10_UV" is very difficult to read in Figure S6b. Please increase font size or delete this label.

186) Both x and y axis descriptions are very difficult to read in Figure S6b. Please increase font size.

187) It is not clear what the authors mean by "Fractions peak" in Figure S6b. Are these the fractions containing the eluted ∆N-FAT-M?

188) It is not clear what is the other dominant peak appearing between around 0 and 65 ml in the chromatogram shown in Figure S6b?

189) In addition to 10, 15, 20, 25, and 37 kDa sizes, please annotate all other molecular weight markers shown in the Coomassie brilliant blue-stained gels present in Figures S6c and S6d.

190) It is not clear what fractions depicted in Figure S6c were used for downstream purification on Superdex-200?

191) It is not clear what are the contaminating bands appearing in Figure S6c and S6d? Please comment on in the figure legend.

192) It is not clear what sample was loaded in the second lane in Figure S6c as it is different from all other lanes?

193) It is not clear what fractions depicted in Figure S6d were used as the final purification product of ∆N-FAT-M?

194) It is not clear at which step was thrombin cleavage performed in Figure S7a?

195) It is not clear how was ∆N eluted from the Ni-NTA and Superdex-200 columns in Figure S7a?

196) The label "Manual Run 0:10_UV" is very difficult to read in Figure S7b. Please increase font size or delete this label.

197) Both x and y axis descriptions are very difficult to read in Figure S7b. Please increase font size.

198) It is not clear what the authors mean by "Fractions peak" in Figure S7b. Are these the fractions containing the eluted ∆N?

199) It is not clear what is the other dominant peak appearing between around 0 and 70 ml in the chromatogram shown in Figure S7b?

200) In addition to 10, 15, 20, 25, and 37 kDa sizes, please annotate all other molecular weight markers shown in the Coomassie brilliant blue-stained gels present in Figures S7c and S7d.

201) It is not clear what fractions depicted in Figure S7c were used for downstream purification on Superdex-200?

202) It is not clear what are the contaminating bands appearing in Figure S7c and S7d? Please comment on in the figure legend.

203) It is not clear what fractions depicted in Figure S7d were used as the final purification product of ∆N?

204) It is not clear at which step was thrombin cleavage performed in Figure S8a?

205) It is not clear how was ∆ATP eluted from the Ni-NTA and Superdex-200 columns in Figure S8a?

206) The label "Manual Run 7:90_UV" is very difficult to read in Figure S8b. Please increase font size or delete this label.

207) Both x and y axis descriptions are very difficult to read in Figure S8b. Please increase font size.

208) It is not clear what the authors mean by "Fractions peak" in Figure S8b. Are these the fractions containing the eluted ∆ATP?

209) It is not clear what is the other dominant peak appearing between around 0 and 35 ml in the chromatogram shown in Figure S8b?

210) Please enlarge Western blot images presented in Figures S8c and S8d so that they roughly match the size of those provided in Figures S3c, S3d, S6c, S6d, S7c, and S7d.

211) In addition to 10, 15, 20, 25, and 37 kDa sizes, please annotate all other molecular weight markers shown in the Coomassie brilliant blue-stained gels present in Figures S8c and S8d.

212) It is not clear what fractions depicted in Figure S8c were used for downstream purification on Superdex-200?

213) It is not clear what are the contaminating bands appearing in Figure S8c and S8d? Please comment on in the figure legend.

214) It is not clear what sample was loaded in the three leftmost lanes in Figure S8c as it is different from all other lanes?

215) It is not clear what fractions depicted in Figure S8d were used as the final purification product of ∆ATP?

216) It is not clear what sample was loaded in the three leftmost lanes in Figure S8d as it contains an extra higher-molecular weight bands?

217) Please indicate how exactly is analyte association and dissociation triggered in the BLItz reaction mixture as part of Figure S9.

Round 2

Reviewer 2 Report

Shams et al. have expanded our understanding of the kinetic landscape of the mammalian/mechanistic target of rapamycin complex 1 (mTORC1) assembly with respect to RHEB-mTOR interaction. The authors have utilized in vitro and cellular binding assays to determine that RHEB interacts with different mTOR domains with distinct affinities while the strongest association was found for RHEB-ΔATP interaction. This is highly relevant since disrupting the mTOR-RHEB axis may interfere with oncogenic signaling. The study is thorough and the manuscript is well written, however a standard discussion section would help readers to better appreciate the significance of the novel kinetic findings perceived in the context of the current knowledge of RHEB/mTOR biology.

Major points:

1) Figure S7d does not seem to be the full gel photo of the cropped image presented in Figure 3e.

2) Figure S8d does not seem to be the full gel photo of the cropped image presented in Figure 3h.

3) Although the authors have aptly indicated that their study was motivated by the need to gain deeper understanding of the molecular mechanisms of cancer in the beginning of the Results section, the manuscript still lacks proper discussion. Please discuss the relevance of the conclusions made in relation to what is already known from previously published research in one or few new paragraphs following right after the results part. In addition, it might be also interesting to speculate what is the reason behind the observed strong RHEB-ΔATP binding?

Minor points:

1) Please incorporate references to Supplementary note 1 and 2 into the main text body.

2) Please change "mTORC1, makes" to "mTORC1 makes" (line 16).

3) It is not clear why the "ΔN-FAT-M" fragment was not constructed as "ΔN-M-FAT" given the order of the respective domains? Please explain in the text.

4) Please replace "with" with "to" (line 23).

5) Please change "at" to "with" (lines 22, 23).

6) Please replace "as of" with "as" (line 23).

7) Please change "the less importance of FAT domain in" to "that the FAT domain is dispensable for" (line 24).

8) "domain" could be replaced with "domain (ΔFAT, aa 1240-1360)" (line 24).

9) Please replace "that of" with "to" (line 25).

10) Please change "binds" to "binds to" or "engages" (line 25).

11) Please replace "to" with "for" (line 26).

12) Please change "probably" to "likely" (line 26).

13) Please replace "(mTORC2),respectively, to" with "(mTORC2) to" (line 33).

14) Please change "actions" to "processes" (line 34).

15) Please replace "to regulate the recruiting" with "which regulates the recruitment" (line 35).

16) Please change "of" to "of the" (line 65).

17) Please replace "turn-on" with "turn on" or "stimulate" (line 70).

18) Please change "RHEB is farnesylated to the lysosomal membrane" to something like "membrane anchoring of RHEB to the lysosomal membrane is mediated by farnesylation" (line 70).

19) "factors and nutrients" could be replaced with "factor and nutrient stimulation" (line 71).

20) Please replace "RHEB-GTP" with something like "RHEB-GTP molecules" or "RHEB-GTP subunits" (line 76).

21) Please change "by the meaning of protein-protein interaction" to "by means of protein-protein interactions" (line 77).

22) Please replace "for the" with "to sustain their" (line 81).

23) Please change "the inhibition has been supposed to work in" to something like "its inhibition has been proposed for" (line 81).

24) Please replace "to develop the anti-cancer drugs, a variety of molecules have targeted to the kinase activity of mTOR" to something like "a variety of molecules have been developed to target the kinase activity of mTOR as anti-cancer agents" (line 82).

25) Please change "mTORC1 specific inhibition turned" to "specific inhibition of mTORC1 turned out" (line 83).

26) Please replace "mTOR inhibition" with "mTOR" (line 84).

27) Please change "inhibit mTORC1 by the inhibitors" to "target mTORC1" (line 85).

28) Please replace "blocking the signal transductions" with "to block signal transduction" (line 85).

29) Please change "For the signal blockade, RHEB is supposed to be" to something like "Accordingly, RHEB represents" (line 86).

30) Please replace "mTORC1 by" with "mTORC1 achieved by" (line 87).

31) Please change "provide insights to develop new anti-cancer drugs based on the blockade of the interaction involved in RHEB" to something like "reveal the kinetics of RHEB binding to mTOR, which can inform the development of new anti-cancer drugs" (line 88).

32) It is not clear what the authors mean by "RT" (line 96) as it stands out separately of any relevant context?

33) Please replace "decrease in the" with "decreased" (line 286).

34) Please change "(different mTOR fragments)" to "representing different mTOR fragments" (line 295).

35) Please replace "suggested the" with "suggested that the" (line 301).

36) Please change "corresponds" to "corresponds with" (line 303).

37) Please change "bound" to "interacted with the" or "engaged the" (line 303).

38) "possible that RHEB regulates the kinase activity of mTOR upon the binding to ΔATP domain" (line 305) is cut through by the white border of Figure 2. Please fix.

39) Please replace "standard deviation (SD)" with "SD" (line 360).

40) Please replace "because of" with "owing to" (line 375).

41) Please change "(See Figure S6D for full gel photo)" to ". See Figure S6d for full gel photo" (line 379).

42) Please replace "270 s:450 s" with "270–450 s" (line 380).

43) Please change "450 s:750 s" to "450–750 s" (line 380).

44) Please replace "(See Figure S7D for full gel photo)" with ". See Figure S7d for full gel photo" (line 381).

45) Please change "of (270 s:450 s)" to "of 180 s (270–450 s)" (line 399).

46) Please replace "450 s:750 s" with "450–750 s" (line 399).

47) Please change "(See Figure S8D for the full gel photo)" to ". See Figure S8d for the full gel photo." (line 400).

48) Please replace "for the" with "for" (line 401).

49) Please change "210 s:330 s" to "210–330 s" (line 401).

50) Please replace "330 s:570 s" with "330–570 s" (line 401).

51) Please change "mTORΔKi" to "mTORΔATP" in the Materials table 2x and in Figure S5e.

52) Please change "inhibitors" to "inhibitor" (line 422).

53) Please replace "by the purification" with "with the purification" (line 425).

54) Please change "LB (Luria-Bertani)" to "Luria-Bertani (LB)" (line 465).

55) Please replace "TB (terrific broth)" with "terrific broth (TB)" (Line 473).

56) It is not clear what do "Y", "Bmax", "X", and "Kd" represent in "Y=Bmax * X/(Kd + X)" (line 676)? 

57) Please change "initiate" to "initiate the" (line 690).

58) It is not clear what does "t" represent in "Y = Y0 + Req (1- e-Kobs*t)" and in "Y = Ye + YΔ e-Kd*t" (line 703)?

59) The sentence "Please refer to the technical notes for further information (https://www.sartorius.com/)" (line 801) seems to be redundant.

60) Please replace "Penicillin / streptomycin" with "Penicillin/streptomycin" in the Abbreviations section.

61) Please change "Illustration created by BioRender (https://app.biorender.com/). (RT: room temperature)" to "RT, room temperature. Illustration created by BioRender (https://app.biorender.com/)" in the legend to Figure S4.

Round 3

Reviewer 2 Report

1) Please replace "While it" with something like "Previously, it" (line 49).

2) Please change "suggesting a" to "suggesting the" (line 111).

3) Please replace "210-330" with "210–330" (line 147).

4) Please change "RHEB, respectively," to "RHEB" (line 153).

5) Please replace "GTP, respectively" with "GTP" (line 158).

6) Please change "the" to "this" (line 163).

7) Please replace "information will contribute to design" with "finding may open new avenues for developing" or "finding will open new avenues for developing" (line 166).

8) Please change "interfering" to "interfering with" (line 167).

9) Please replace "Actually" with "In fact" (line 168).

10) Please change "this study will inform us to develop new inhibitors for mTORC1, based on the kinetics of RHEB obtained" to "the kinetics of mTORC1-RHEB binding obtained in this study will inform the future development of new inhibitors of mTORC1" or "the kinetics of RHEB binding to mTORC1 obtained in this study will inform the future development of new inhibitors of mTORC1" (line 169).

11) Please replace "then," with "then" (line 180).

12) Please change "w/v" to "(w/v)" (line 190).

13) Please replace "membranes" with "membrane" (line 192).

14) Please change "silicon-type" to "silicone-type" (line 207).

15) Please change "1mM" to "1 mM" (line 209).

16) Please replace "cycle / sec" with "cycle/s" (line 216).

17) Please change "AKTAprime" to "ÄKTAprime" (line 218).

18) Please replace "elusion" with "elution" (lines 219, 230).

19) Please change "so, the" to ", so the" (line 223).

20) Please replace "PBS, respectively" with "PBS" (line 241).

21) Please change "tri-phosphate" to "triphosphate" in the Abbreviations section.

22) Please change "(b) Whole view of Western blot analysis of the different stages of mTOR purification showing the expression of the Halo-tagged mTOR. TEV site (EDLYFQ↓S) is a cleavage site for TEV protease" to "TEV site (EDLYFQ↓S) is a cleavage site for TEV protease. (b) Whole view of Western blot analysis of the different stages of mTOR purification showing the expression of the Halo-tagged mTOR" (legend to Figure S2).

23) Please replace "from the protein oligomerization" with "from protein oligomerization" (legend to Figure S6).